# Updates on Molecular Targeted Therapies for Intraparenchymal CNS Metastases

**DOI:** 10.3390/cancers14010017

**Published:** 2021-12-21

**Authors:** Akanksha Sharma, Lauren Singer, Priya Kumthekar

**Affiliations:** 1Department of Translational Neurosciences, Pacific Neuroscience Institute, Saint John Cancer Institute, Santa Monica, CA 90404, USA; 2Malnati Brain Tumor Institute at the Robert H. Lurie Comprehensive Cancer Center, Department of Neurology at the Feinberg School of Medicine, Northwestern University, Chicago, IL 60611, USA; lauren.singer@nm.org (L.S.); priya.kumthekar@nm.org (P.K.)

**Keywords:** intraparenchymal metastases, CNS disease, metastatic disease, targeted therapy, immunotherapy, tyrosine kinase inhibitors, monoclonal antibodies

## Abstract

**Simple Summary:**

Metastatic disease to the central nervous system is an advanced-stage complication with historically devastating consequences and high mortality. Significant progress has been made in treatment in the last two decades, especially with the identification and targeting of specific mutations in the cancer pathway. In this review, we provide an updated overview of specific targets and highlight the numerous drugs that have demonstrated penetration and efficacy within the central nervous system.

**Abstract:**

Central nervous system (CNS) metastases can occur in a high percentage of systemic cancer patients and is a major cause of morbidity and mortality in these patients. Almost any histology can find its way to the brain, but lung, breast, and melanoma are the most common pathologies seen in the CNS from metastatic disease. Identification of many key targets in the tumorigenesis pathway has been crucial to the development of a number of drugs that have demonstrated successful penetration of the blood–brain, blood–cerebrospinal fluid, and blood–tumor barriers. Targeted therapy and immunotherapy have dramatically revolutionized the field with treatment options that can provide successful and durable control of even CNS disease. In this review, we discuss major targets with successful treatment options as demonstrated in clinical trials. These include tyrosine kinase inhibitors, monoclonal antibodies, and antibody–drug conjugates. We also provide an update on the state of the field and highlight key upcoming trials. Patient-specific molecular information combined with novel therapeutic approaches and new agents has demonstrated and continues to promise significant progress in the management of patients with CNS metastases.

## 1. Introduction

Metastatic cancer can often find its way to the brain, where deposits may form either in the brain parenchyma itself resulting in intracranial or intraparenchymal metastases (IPM) or colonize the cerebrospinal fluid (CSF) surrounding the brain and spinal cord, resulting in leptomeningeal disease (LMD). Central nervous system (CNS) spread of systemic cancer as IPM or LMD is estimated to occur in 5–40% of patients with metastatic cancer; however, the actual prevalence may be even higher given CNS spread is not always identified before death and not routinely reported to state cancer registries [1,2]. Lung, breast, and melanoma are the most common sources of CNS metastases, though any cancer may metastasize to the parenchyma or CSF. IPM result in significant morbidity and negatively impact median overall survival (OS); indeed, patients with IPM are considered to have late or advanced stage cancer with a survival typically estimated to be less than six months [3]. Radiation therapy (RT), either via stereotactic radiosurgery (SRS) or whole brain radiation therapy (WBRT), remain the primary modalities of treatment. However, there has been a notable increase in systemic therapy options for patients with IPM over the last decade, which has dramatically improved the landscape in terms of both progression-free survival (PFS) and OS for patients with several of these cancers.

Systemic options that have been more successful in controlling intracranial and extracranial disease are those that specifically target genomic alterations in the tumor. Several actionable genetic alterations have been identified in a range of primary cancers. In this review, we aim to discuss the most common and significant mutations and their respective targeted therapies. Figure 1 provides a visual overview of these targets and highlights the key drugs currently available that can target these mutations to inhibit downstream signaling pathways and also have been noted to have some degree of penetration and efficacy in the CNS. It is important to note, however, that IPM may not always share the same alterations as the extracranial disease. Genetic makeup of the primary cancer is not necessarily always a surrogate for the alterations that may be seen within CNS disease through a phenomenon called “branched evolution,” suggesting the need for sampling directly from the CNS when feasible [4,5].

## 2. ALK-Targeted Therapies

The anaplastic lymphoma kinase (ALK) gene translocation is noted in 4–7% of non-small cell lung cancer (NSCLC) cases and results in a fusion between ALK and a second gene (most commonly EML4). ALK is a key regulator of tumor cell growth and survival, and this translocation results in increased activation of the signaling pathway, promoting oncogenic cell proliferation and survival. The tyrosine kinase domain of ALK can be targeted by a number of tyrosine kinase inhibitors (TKIs) (Figure 1). Crizotinib was the first of this class of drugs but demonstrated only marginally improved intracranial activity compared to chemotherapy. The newer generations of ALK inhibitors including ceritinib, alectinib, brigatinib, lorlatinib all demonstrated greater blood–brain barrier (BBB) penetration and CNS activity. Phase III trials in NSCLC with ceritinib have demonstrated an improved PFS when compared to chemotherapy (5.4 ms vs. 1.6 ms) [6]. In a phase II trial with pre-treated NSCLC patients, median PFS was 16.6 months and median overall survival (OS) was 51.3 months [7]. Intracranial disease control rate (DCR) was as high as 80% with a median duration of response (DOR) of 24 months [7]. A trial with leptomeningeal disease (LMD) from NSCLC also demonstrated an overall response rate (ORR) of 16.7% with OS of 7.2 months in the LMD group [8].

Alectinib similarly demonstrates CNS activity and PFS benefit in patients regardless of IPM status. When compared to crizotinib, alectinib demonstrates a significantly high PFS (not reached vs. 10.2 months) [9]. In addition, alectinib has been shown to be protective against CNS disease progression based on results from a Phase III study in which only 12% in the alectinib arm had intracranial disease progression versus 45% in the crizotinib arm [10]. Alectinib generally was well tolerated, with primary side effects being anemia, myalgias, weight gain, and photosensitivity. Crizotinib, on the other hand, has a higher rate of nausea, diarrhea, and vomiting [10].

Brigatinib similarly demonstrates a better profile when compared to crizotinib and appears to be well tolerated. In a trial involving patients with NSCLC, median PFS was 29 months with brigatinib versus 9.2 months with crizotinib, with a confirmed rate of intracranial response rate of 78% vs. 29%, respectively [11]. Diarrhea is more common with brigatinib than alectinib, and other side effects included elevated creatine phosphokinase, cough, hypertension, and increased liver function tests [11].

Lorlatinib is a third generation TKI that has been designed to cross the BBB. In a phase III trial comparing lorlatinib to crizotinib that enrolled untreated patients with ALK rearrangements, intracranial response was 66% vs. 20%. As many as 71% of patients were noted to have complete response (CR) intracranially and at 12 months 72% still maintained response suggesting impressive durability to treatment. Similar to alectinib, lorlatinib tends to delay time to CNS progression, with the risk of CNS progression as low as 3% with lorlatinib versus 33% with crizotinib [12]. Lorlatinib is noted to have an added risk of memory impairment and cognitive issues.

Given the robust response data seen even in untreated patients with these later generation TKIs, the question arises if radiation therapy (RT) should be deferred or included for IPM from ALK rearranged NSCLC. No prospective data is available, and retrospective studies still suggest that there is benefit of added RT [13]. In specific clinical scenarios, including patients with small or asymptomatic IPM, IT may be reasonable to defer upfront RT for systemic therapy first.

ALK rearrangements are generally mutually exclusive to the other mutations discussed here with the exception of ROS1, which may co-exist with the ALK translocation and is discussed separately in this review. It is rare now in most countries where these drugs are available to use standard chemotherapy as first-line therapy and for patients with known IPM or relapsed/progressive disease with IPM, we recommend the use of lorlatinib or brigatinib to achieve disease control given the increased CNS penetration and excellent demonstrated efficacy as discussed above. Careful consideration of individual patient tolerance and risk of side effects should also be part of the decision-making process.

## 3. EGFR Targeted Therapies

The epidermal growth factor receptor (EGFR) is a member of the ErbB family of receptors. This transmembrane protein has important activity that can encourage growth factor signaling—over-expression or activation of the EGFR pathway results in increased cell proliferation and cell survival, via downstream activation of the phosphatidylinositol-3-kinase (PI3K/AKT) and Janus kinase (JAK/STAT) pathways. This mutation has been noted to occur in up to 35% of primary NSCLC patients, with a higher rate in those with an Asian ethnicity. The third-generation drug osimertinib is especially effective as a TKI for EGFR especially given it can also target the T790M mutation, an escape mutation on exon 20 that has been seen to confer resistance to TKI therapy. Osimertinib has demonstrated efficacy in treating EGFR-mutant NSCLC with CNS extension when compared to chemotherapy (platinum/pemetrexed) and to previous generation TKIs (gefitinib or erlotinib), a situation which prior to this would have had few therapeutic options. In the AURA 3 trial, osimertinib was compared to the previous standard chemotherapy (a combination of platinum/pemetrexed), and the CNS overall response rate was 70% vs. 31%. Median CNS response duration was noted to be 8.9 months [14]. When osimertinib was compared to gefitinib or erlotinib in the FLAURA trial, osimertinib demonstrated a CNS objective response rate of 91% and a median PFS that was not reached vs. 13.9 months in the control arm [15]. New CNS lesions only occurred in 12% of the osimertinib arm vs. 30% of the control arm, also suggesting a protective effect, with an overall median OS of 39 months vs. 32 months [15,16]. For LMD, a phase II prospective study found an impressive intracranial response rate of 55% and a median OS of 16.9 months for NSCLC with LMD. Osimertinib is generally well tolerated, with the most common side effects being diarrhea, dry skin, rash, and mucositis.

Osimertinib monotherapy is therefore becoming the standard first line therapy for EGFR mutated lung cancer. Inclusion of RT, specifically SRS, is also being questioned. While SRS may help with drug penetration or sensitize existing IPM, there is no clear randomized data to support this currently. Previous retrospective studies looked at this question with previous generation TKIs and found that addition of SRS did appear to improve survival [17]. Osimertinib is notably superior to these previous generations, however, in terms of IC response rate, and retrospective data demonstrates that RT may not add much benefit [18]. An ongoing prospective trial evaluating osimertinib versus osimertinib with SRS aims to better answer this question (NCT03769103, Table 1).

On the horizon is tesevatinib, a novel TKI with selectivity towards both EGFR and vascular endothelial growth factor (VEGF) that has demonstrated promising CNS penetration [19]. A phase II clinical trial in NSCLC brain metastases is evaluating this drug (NCT02616393, Table 1).

EGFR mutations are noted in other solid cancers such as colon cancer, esophageal cancer, glioblastoma, etc. However, at this time, studies utilizing EGFR TKIs in these other pathologies have not demonstrated the same level of efficacy or success in arresting tumor growth (especially when it comes to the CNS) as what has been seen in NSCLC. In our practice, the development of osimertinib has truly changed the landscape for patients with EGFR-mutant NSCLC, allowing for a prolonged period of disease remission even with CNS IPM, with relatively tolerable side effects. Osimertinib may also be used in the setting of small and asymptomatic brain metastases where RT is being deferred.

## 4. ROS-1 Alterations

A rare alteration, seen in only 1–2% of NSCLC, ROS1 is a receptor tyrosine kinase that is downstream of the c-ros oncogene. This rearrangement is similar to that of ALK and is seen also in glioblastoma, cholangiocarcinoma, ovarian carcinoma, angiosarcomas, etc. Aberrant ROS1 can activate multiple oncogenic pathways downstream, thus leading to tumor proliferation and survival.

It is noted that in NSCLC, a ROS1 fusion mutation predicts better response to pemetrexed based therapy, an agent which has been known to have CNS penetration [20,21]. Amidst the TKIs, crizotinib has been evaluated in the NSCLC population and trials have included IPM [22]. Median PFS for those with IPM was 10.2 months, and 13.8 months for those without IPM [23]. Lorlatinib has a higher potency against ROS1 and as discussed previously has excellent BBB penetration. An early phase study has demonstrated response intracranially in three patients with ROS1 mutated IPM but additional studies are ongoing (see Table 1) [22]. Entrectinib, discussed in the next section, may also be used to treat ROS1 fusion NSCLC. Anecdotal evidence and case reports suggest that other pathologies may also respond to these drugs or other ROS1-specific targeted TKIs, but additional data is needed and trials are ongoing at this time.

## 5. NTRK

Neurotrophic tyrosine receptor kinase or NTRK gene fusions can be seen in colorectal cancer, NSCLC, cholangiocarcinoma, glioblastoma, sarcoma, and thyroid cancers, amidst others. They involve NTRK1, 2 OR 3, which encode for the respective neurotrophin receptors (TRKA, TRKB, TRKC) and in turn this activation leads to oncogenesis. Entrectinib and repotrectinib are TKIs with affinity for these tyrosine receptor kinases (TRKs) and CNS penetration [24,25]. A pooled analysis of entrectinib in patients with NSCLC who had NTRK1 and ROS1 mutations demonstrated that 11 of 20 patients (55%) with baseline CNS metastases had a response, with median DOR of 12.9 months. Median intracranial PFS was 7.7 months [26,27]. A recent updated analysis of NCT02576431 and NCT021122913 presented this year demonstrated that heavily treated patients with advanced lung cancer and known IPM demonstrated an overall response rate to larotrectinib of 63%. Twelve-month PFS was 65% and median OS was 40 months, which is quite encouraging [28]. The drug was tolerable, with the most common side effects being fatigue, dysgeusia, paresthesias, nausea, and myalgias [27,28]. Additional larger trials are being conducted in other solid cancers that may carry this mutation, including glioblastoma. For NTRK and ROS mutated lung cancer, consideration of this class of drugs is highly advised in clinical practice both in the post RT setting as well as in the small and asymptomatic brain metastases setting where deferring RT may be preferred.

## 6. KRAS

The Kirsten rat sarcoma viral oncogene homolog or KRAS gene is aberrant in NSCLC (up to 25% of cases), colorectal cancers, and pancreatic ductal adenocarcinomas. An activating mutation in the KRAS gene results in increased formation of the K-Ras protein, a notable part of the RAS/MAPK pathway (Figure 1). This protein provides signals for cells to grow and proliferate, thus contributing to tumorigenesis. Until recently, the KRAS mutation was noted to be a poor prognostic indicator due to the lack of targeted options available and the fact that it appears to drive resistance to EGFR inhibition [29]. In recent years, however, more exciting options have emerged that suggest that KRAS inhibition is possible. Sotorasib was examined in advanced solid tumors that included NSCLC and colorectal cancers that had failed multiple lines of treatment. There was an objective complete response of 32% noted. This trial included patients with IPM though that subset has not been separately reported yet, but this holds promise for the future [30]. Other drugs being investigated in solid tumors include combinations with selumetinib or binemetinib, drugs that do have CNS penetration. This will be an area that will hold continued interest in the coming years, both for NSCLC and for other solid tumors that might also have the KRAS mutation.

Of note, immunotherapy with pembrolizumab demonstrates response in NSCLC regardless of KRAS status. When compared to chemotherapy, patients on pembrolizumab had a response rate of 57% (vs. 18%) in the KRAS subgroup of a larger trial [31]. This drug does have CNS penetration and activity against IPM as discussed in another section.

## 7. CDK4/6

The activation of cyclin-dependent kinases CDK4 and CDK6 in several cancers leads to increased, unregulated cell proliferation. Inhibiting these kinases can lead to cell cycle arrest and apoptosis of tumor cells. Currently, there are three FDA-approved CDK4/6 inhibitors—palbociclib (inhibits both CKD4 and CDK6), ribociclib (similar to palbociclib in structure but more potent against CDK4), and abemaciclib (different in structure and more potent against CDK4 also) [32]. These drugs have demonstrated efficacy and survival benefit in hormone positive breast cancer but intracranial response and benefit remains unclear and yet to be explored. Abemaciclib has better CNS penetration and early efficacy for IPM has been demonstrated with a phase II study demonstrating an intracranial benefit rate of 24% specifically for patients with HR+, HER2 negative, previously treated IPM [33]. Importantly, in this study, abemaciclib achieved therapeutic concentrations in the tissues of IPM, beyond what is required for CDK4 and CDK6 inhibition. The drug appears safe and is well-tolerated with mainly gastrointestinal side effects. Currently, additional evidence is being gathered in IPM specific clinical trials, but at this time it is clinically used for breast cancer with CNS spread (NCT03994796, Table 1).

## 8. Her2+ Targeted Therapies

The HER2 membrane tyrosine kinase is a member of the epidermal growth factor receptor family. Overexpression and gene amplification is an aberrancy noted in several solid cancers including breast, esophageal, ovarian, colorectal, etc. The upregulated expression of HER2 leads to downstream signaling pathway activation, thus leading to cell growth and proliferation, and preventing cell death. HER2 is noted to be upregulated in IPM when compared to the systemic disease, which explains the increased risk of HER2 tumors of colonizing the CNS. Small molecular TKIs including lapatinib, neratinib, and tucatinib have shown to have intracranial benefit in IPM from breast cancer, but only when used as combination therapy with capecitabine, with or without trastuzumab. Lapatinib combined with capecitabine demonstrates relatively low toxicity as well as an intracranial response rate of 38% with a PFS of 5.5 months in metastatic breast cancer to the brain [34]. Neratinib plus capecitabine has been compared to lapatinib plus capecitabine and the former demonstrated a higher PFS of 7.8 months with a combined intracranial response rate of 35% in the same population [35]. Tucatinib, when combined with both capecitabine and trastuzumab, has demonstrated the highest efficacy in reducing intracranial progression, with an intracranial response rate as high as 50% in metastatic breast cancer patients already previously treated with pertuzumab/trastuzumab [36,37]. Phase I studies have also demonstrated that even without capecitabine, tucatinib and trastuzumab combined results in a successful intracranial response and a clinical benefit (in patients with breast cancer previously treated with trastuzumab and ado-trastuzumab emtansine) [38,39]. This combination may also benefit patients with LMD, and this is being further explored in clinical trial (NCT03501979, Table 1).

Pyrotinib is a newer TKI that has been evaluated in patients with IPM with promising results. In a small cohort of 39 patients with IPM from breast cancer, median PFS was 8.7 ms and OS was 14 ms, with a response rate of 24% [40]. A similar response rate was seen in a prospective analysis from China, where intracranial response rate was 28% in previously treated breast cancer patients [41]. A similar response rate of 25% has been noted in patients with the more rare group of patients with HER2+ NSCLC treated with pyrotinib monotherapy [42]. Radiotherapy-naïve breast cancer patients with IPM were evaluated in a phase II trial where CNS response rates with pyrotinib in combination with capecitabine were noted to be as high as 75% and median PFS was 12.2 months, higher than the group that had been treated with RT [43]. Patients included in this study were required to be TKI naïve, and therefore while there is likely a role for pyrotinib in IPM, the appropriate sequencing with regards to other TKIs needs further clarification.

HER2 can also be targeted by monoclonal antibodies that have traditionally been considered to be unable to traverse the BBB. However, preclinical studies have demonstrated that at higher doses, trastuzumab does have BBB penetration [44,45]. This work provided the foundation of the PATRICIA study, evaluating high dose pertuzumab and trastuzumab together in patients with IPM from breast cancer [44]. This therapy was generally well tolerated and while the primary endpoint was not met due to a modest overall response rate (11%), the clinical benefit rate for these predominantly pretreated patients was 68% at 4 months and 51% at 6 months [44]. With all of this data in mind, at this time, our clinical practice recommendation is to consider the use of a TKI (most commonly tucatinib) with pertuzumab or trastuzumab, and capecitabine, in patients presenting with IPM to the brain from breast cancer. RT still has a critical role in IPM from breast cancer, and these patients may also receive combination SRS and/or WBRT in most cases for intracranial disease, at least until additional data shows non-inferiority of these treatment regimens.

HER2 targeted monoclonal antibodies may be conjugated to drugs (antibody—drug conjugates, or ADCs) to increase CNS penetration and efficacy. Trastuzumab conjugated to emtansine (T-DM1) is one such agent that was evaluated in the KAMILLA single arm phase IIIb trial. Patients with previously treated metastatic breast cancer were enrolled and in the IPM subgroup the median PFS was 5.5 months with an OS of 18.9 months, and an intracranial response rate of 21% [46]. T-Dxd or trastuzumab deruxtecan is an ADC that combines a topoisomerase I inhibitor to trastuzumab and is FDA-approved for patients with HER2+ advanced breast cancer after ≥2 lines of systemic therapy, based on data from the DESTINY-Breast01 phase 2 trial. Although patients with active, symptomatic IPM were excluded, those with asymptomatic IPM demonstrated a response rate of 41% and median PFS of 18 months, showing activity in the brain [47].

Trastuzumab may also be utilized intrathecally for patients with HER2 positive LMD. Doses ranging from 30 to 150 mg have been explored in phase I and II studies with no dose limiting toxicities and improvement in survival and clinical response as compared to historical controls [48,49,50]. A phase II study is ongoing (NCT01373710). Intrathecally delivered trastuzumab is not thought to have the same impact on parenchymal brain metastases and therefore its use is currently limited to the LMD setting BRAF inhibitors.

The most common BRAF mutations include the V600E substitution (valine substituted for glutamic acid) or the V600K mutation (valine substituted for lysine). As a consequence of these mutations, the MAPK pathway is upregulated, and cell cycle proliferation is encouraged. This mutation is most common in melanoma, where 50% of IPM might harbor a BRAF mutation. BRAF inhibitors include vemurafenib, dabrafenib, and encorafenib. Vemurafenib can have CNS efficacy as monotherapy, with a phase II study demonstrating response rates of 20% for treated and untreated IPM [51]. Dabrafenib monotherapy in the BREAK-MB trial demonstrated an intracranial response rate of 39% in BRAFV600E mutated IPM from melanoma; V600K mutated tumors had a lower response rate [52,53]. Combining MEK inhibition aides in overcoming drug resistance and improves the efficacy of BRAF inhibition, and thus the COMBI-MB trial combined dabrafenib with trametinib in patients with BRAFV600 mutant IPM. Intracranial responses as high as 58% were seen in these patients, which included cohorts of previously treated (with RT) and untreated patients [54]. At this point, BRAF therapy is a routine part of metastatic melanoma care and has dramatically changed the landscape in terms of PFS and OS for these patients, including for those with IPM. Combinations with concurrent immunotherapy, as well as the benefit of RT in this population, are questions still undergoing investigation. BRAF therapy may be utilized both in the post brain RT setting as well as can be a very reasonable treatment option for small and asymptomatic brain metastases without RT.

## 9. PD-1/PD-L1 Inhibition

Monoclonal antibodies targeting the programmed cell death protein 1 (PD-1) receptor, or its ligand (PD-L1), have increasingly emerged as a highly efficacious treatment for several cancers, including lung and melanoma. A tumor cell that overexpresses PD-L1 is able to attract PD-1 and thus protect itself from the body’s own cytotoxic T-cell mediated immune mechanism which would kill aberrant and proliferating cells. Antibodies that inhibit this process by targeting either the protein or the ligand can boost the immune response against these tumor cells. A number of these checkpoint inhibitors have been approved in recent years and many others are being investigated. Nivolumab and pembrolizumab are the two most utilized PD-1 inhibitors, while atezolizumab, avelumab, and durvalumab are gaining prominence as PD-L1 inhibitors. Nivolumab has been combined with an antibody against the cytotoxic T-lymphocyte-associated protein 4 (CTLA-4) receptor, ipilimumab, to increase the immune response generated against cancer cells.

The phase II Checkmate 204 study combining nivolumab and ipilimumab recently released five-year follow up data. This trial included asymptomatic melanoma IPM and at 36 months OS had not yet been reached for 72% of patients, which demonstrated both the efficacy and durability of this response. An intracranial response rate of 55% was noted [55,56]. Of note, neurologically symptomatic patients and those already on steroids did not appear to glean significant benefit from this treatment combination. The ABC study from Australia was also a phase II trial that included cohorts with and without prior brain therapy. Again, intracranial response rate was high at 59%. Patients who had IPM that were previously treated, and those with LMD, responded less than those with untreated IPM [52].

BRAF inhibition may be combined with these monoclonal antibodies in patients with melanoma who have both PD-1 positivity and BRAF mutations, but trial data for this combination treatment is still pending at this time. There also remains question on the benefit of RT in these patients—radiation may provide increased durability to response and retrospective data suggests better survival and lower rate of CNS progression, but this has not yet been demonstrated prospectively [57,58,59] There also may be a higher rate of radiation necrosis and unnecessary toxicity in these patients that can be compounded by the use of immunotherapy, the rate of this complication is variable but may be as high as 15–20% [59,60].

Pembrolizumab, another PD-1 inhibitor, when combined with chemotherapy for NSCLC patients with IPM provides a notably higher clinical benefit, with a response rate of 39% (vs. 19.7% for chemo alone) and a durable response with a median OS of 18 months vs. 7.6 months [61]. These monoclonal antibodies have been overall very instrumental in transforming the landscape for patients with melanoma and lung cancer, completely changing survival even with advanced stage cancer with IPM. Their utility is not limited to these cancers alone—in fact, immunotherapy is rapidly integrating into regimens for a number of solid cancers including gastric, bladder, head and neck, esophageal, squamous cell, etc., resulting in higher rates of survival and improved outcomes for a large percentage of cancer patients. Immunotherapy is not without toxicity, of course, and patients are at risk for immune-mediated complications such as skin rashes, pneumonitis, colitis, hepatitis, and may have life-threatening or heavily disabling neurological complications. The data supports immunotherapy to be used for specific primary histologies (i.e., NSCLC, melanoma) even in the absence of RT particularly for small and asymptomatic brain metastases.

## 10. Other Agents

Another ADC composed of an antibody targeting the trophoblast cell-surface antigen 2 (Trop 2) coupled with a topoisomerase I inhibitor govitecan led to the development of sacitizumab govitecan (not included in Figure 1). Recently, results from a randomized phase 3 trial comparing sacituzumab govitecan to single agent chemotherapy in relapsed and refractory triple negative breast cancer were reported, demonstrating promising response with a median OS of 12.1 months compared to 6.7 months [62]. This trial, however, excluded IPM. A separate study, ASCENT 3, did allow for stable asymptomatic IPM and intracranial response rate for the sacituzumab govitecan group was 3% vs. 0% with chemotherapy [63]. Additional clinical trials evaluating sacituzumab govitecan in brain metastases are ongoing (NCT04647916).

BRCA1 and 2 can be targeted by PARP inhibitors such as olaparib and talazoparib, but intracranial response rate in active IPM is still to be explored and reported. The phase III EMBRACA trial with talazoparib did include a subgroup of treated and stable IPM patients who appeared to still benefit in terms of PFS [64]. An ongoing trial with veliparib is aiming to further answer this question (NCT02595905). At this time, additional data is awaited to make additional assessments on the utility of these therapies for patients with known IPM.

Medical therapy is often pursued after patients progress after standard of care radiation therapy and if there are no other targeted or immunotherapy options available. In a Phase II study enrolling solid tumor IPM patients who progressed following WBRT, patients were treated with bevacizumab at a dose of 10 mg/kg IV every two weeks until CNS disease progression. Response rate was 25% and the 6-month PFS: 46% (95% CI: 25–67%) and median PFS was 5.3 months. Median OS was 9.5 months (95% confidence interval 6.3 m–15.0 m) and QOL was maintained through treatment and there was no noted central nervous system bleeding. Of the 24 evaluable patients, 81% (22/24) experienced clinical benefit defined as stable disease or better [65]. Bevacizumab also may have a notable role in treating radiation necrosis from SRS in patients with IPM who cannot tolerate steroids due to side effects or where the necrosis and edema is proving to be steroid-refractory [66,67].

## 11. Conclusions/Future Directions

Systemic advancements over the past decade in oncologic care have led to improved outcomes for solid tumor cancer patients. Despite these advancements, the incidence of IPM continues to increase as patients live longer and as many of the currently utilized therapeutics do not cross the blood—brain barrier. The development of novel compounds including targeted therapies, ADCs, and immunotherapy amongst other advancements including more sophisticated imaging techniques have brought CNS metastases to the center stage. While there have been improvements in patient outcomes with these advents, there is still much more to understand and explore, and many unanswered question. This is in-part due to the lack of inclusion of patients with active IPM in the key clinical trials which have led to regulatory approval of many of these agents as well as inspired the design of additional studies. Given the only increasing incidence of IPM, it is crucial that these patients be included in clinical trials as they reflect the true populations seen in oncology clinics across the world. While trial design is challenging in this population, the incidence of IPM is 10-fold that of primary brain tumors, and as such should be given appropriate spotlight. This focus will ideally lead to better outcomes for our IPM patients across all primary tumor histologies.

## Figures and Tables

**Figure 1 cancers-14-00017-f001:**
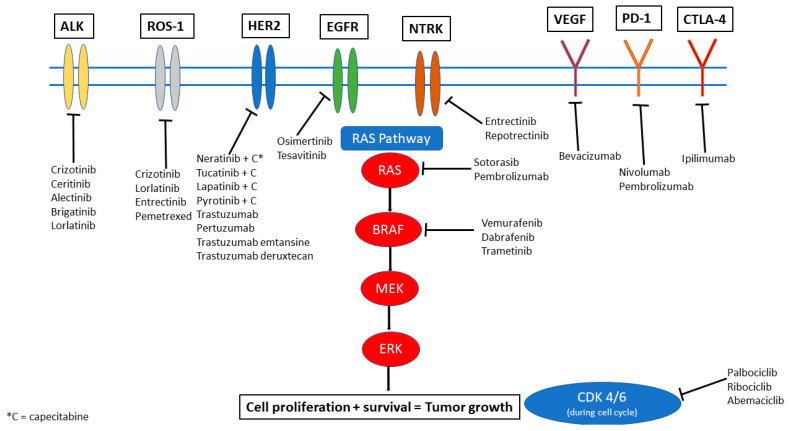
Therapeutic options illustrated by molecular target.

**Table 1 cancers-14-00017-t001:** Ongoing trials targeting IPM with targetable mutations.

Targeted Mutations	Trial	Phase	Population	Investigational Drug(s)	Total *n*	Primary Outcome	Comments
ALK, ROS1	NCT02927340	II	NSCLC	Loratinib	30	Intracranial disease control rate	
ALK, ROS1	NCT01970865	I/II	NSCLC	PF-06463922 vs. Crizotinib monotherapy	334	Participants with DLT, percentage of participants with overall and intracranial ORR	PF-0643922—ALK/ROS1 inhibitor
ALK, ROS1, or NTRK1-3	NCT03093116	I/II	Any IPM	Repotrectinib	450	DLT, recommended Phase II dose, ORR	Multiple arms comparing prior TKI and/or chemotherapy and treatment naïve
ALK, ROS1, NTRK1-3	NCT05004116	I/II	Any IPM	Repotrectinib + Irinotecan + Temozolomide	50	Incidence of DLT, MTD	
EGFR	NCT03769103	II	NSCLC	SRS + Osimertinib vs. Osimertinib monotherapy	76	Intracranial PFS	Treatment naïve brain mets included
ROS1	NCT04621188	II	NSCLC	Loratinib	84	ORR	Recurrence after failure of first-line TKI
ROS1	NCT03612154	II	NSCLC	Loratinib	35	ORR	
ROS1	NCT04919811	II	NSCLC or other IPM	Taletrectinib (DS-6051b)	119	ORR	
ROS1, NTRK	NCT02675491	I	Any IPM	DS-6051b	15	Number and severity of adverse events	
CDK, PI3K, NTRK/ROS1	NCT03994796	II	Any IPM	Abemaciclib or Paxalisib or Entrectinib	150	ORR	CDK population—Ademaciclib, PI3K—Paxalisib, NTRK/ROS1—Entrectinib
KRAS, EGFR	NCT01859026	I/IB	NSCLC	Erlotinib + MEK162	43	MTD	
KRAS	NCT03299088	I	NSCLC	Pembrolizumab + Trametinib	15	Incidence of DLT	
KRAS	NCT03170206	I/II	NSCLC	Palbociclib or Binimetinib monotherapy vs. combination therapy	72	MTD, safety and tolerability, PFS	CDK4/6 inhibitor + MEK inhibitor
KRAS	NCT03808558	II	NSCLC	TVB-2640	12	Disease control rate and response rate	
KRAS	NCT04111458	I	Any IPM	BI-1701963 monotherapy vs. co-administration with Trametinib	80	MTD based on DLT, number of patients with DLT, ORR	
KRASG12C	NCT03785249	I/II	Any IPM	MRTX849 (Adagrasib) monotherapy vs. combination therapy with Pembrolizumab, Cetuximab, or Afatinib	565	Safety, pharmacokinetics, and clinical activity/efficacy of MRTX849	
CDK	NCT02896335	II	Any IPM	Palbociclib	30	Clinical benefit rate (intracranial)	
HER-2 negative	NCT04647916	II	Breast cancer	Sacituzumab Govitecan	44	ORR	
BRAFV600	NCT03911869	II	Melanoma	Encorafebib + Binimetinib vs. high dose	13	Incidence of DLT, incidence and severity of AE, incidence of dose modifications and discontinuations due to AE, brain metastasis response rate	
Checkpoint inhibition	NCT03340129	II	Melanoma	Ipilimumab + nivolumab w/ RT vs. Ipilimumab + Nivolumab alone	218	Neurological specific cause of death	

AE: adverse effects, DLT: dose-limiting toxicity, IPM: intraparenchymal metastases, MTD: maximum tolerated dose, NSLC: non-small cell lung cancer, ORR: overall response rate, PFS: progression free survival, TKI: tyrosine kinase inhibitor.

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
