# Peer review of "Updates on Molecular Targeted Therapies for Intraparenchymal CNS Metastases"

_cancers, 2021, doi:10.3390/cancers14010017_

Round 1
Reviewer 1 Report
The manuscript is interesting and as a review it covers many topics.
From a stylistic point of view, several times words are cut by a hyphen, please check it all through the manuscript (e.g. line 51 ge-nomics, line 66 in-cluding etc).
Regarding the concept of the review, often I felt that an expert opinion on the investigation of the different targeting was missing. Please consider adding considerations on advantages or features of targeting these pathways in each paragraph, or a single paragraph at the ending, in order for the reader to be able to critically distinguish among all these molecules.
Reviewer 2 Report
In this review are discussed the latest therapies opportunity and their targets as demonstrated in clinical trials for cancers leading to central nervous system (CNS) metastases. These include tyrosine kinase, inhibitors, monoclonal antibodies, and antibody-drug conjugates that are listed in separate section in the manuscript. The manuscript is interesting for scientists not in this field of study as well as a short compendium of the list of treatment in trials and on the current state of art. However I would add a chapter with more information on the pathologies that can be treated with these therapies, as this would increase the interest of the readers and the challenges that these treatment appear to successfully overcome.
All together the manuscript is well written and I support its publication with the addition of comments on the molecular bases of drug efficacy in a given disease and reasons) why these therapies improve current state of art.
Minor points :
Line 110: Erlo-tinib =Erlotinib
Line 114 Demon-strated=Demonstrated
Line 86: El-eveted =Eleveted
Line 116: pro-tective=protective
Line 125: im-proved =improved
Line 127: pro-spective =prospective
Reviewer 3 Report
This review is an updated version of the latest findings on molecular-targeted therapeutics and contains valuable information on monoclonal antibody drugs based on molecular targets. It has reached the level of an acceptable review if some problems are corrected.
Minor points
1 Figure: The author should add the title and the legend of the figure as Figure 1.
2 Figure: The figure should be listed in the order of appearance in the text, although it is categorized by headline items in the text. Please take this into consideration for the reader.
3 Table: Table 1 is categorized by trial, but it should be arranged in the order of appearance of molecular targets and drugs in the text.
Please arrange them in the order of appearance of the molecular targets and drugs in the text. For example, starting from the left column, the desired order is the molecular target, drug, population, number of population, phase, outcome, trial, and Note.
4 There are drugs in the text that are not listed in the Figure. Please either include them in the figure or clearly indicate that they are not included in the figure.
5 Is Bevacinumab not included in Other Agents? Other agents are preferred before the conclusion.
6 P3 Line77: Phase 3 should be corrected to Phase III.
Round 2
Reviewer 1 Report
I can see the extent of the work that has been done by authors in response to reviewers' requests. I support the publication of this manuscript.
This manuscript is a resubmission of an earlier submission. The following is a list of the peer review reports and author responses from that submission.